# Applying Action Research in Design Curricula to Fulfill University Social Responsibility—A Case Study of the Pnguu Community of the Tsou Tribe

**Hsiu Ching Laura Hsieh**

Department of Creative Design, National Yunlin University of Science and Technology, Yunlin 64002, Taiwan; laurarun@gmail.com; Tel.: +886-5-5342-601 (ext. 6423)

**Abstract:** Due to the mainstream emphasis on university social responsibility (USR) in global and Taiwanese higher education, there is a growing need for universities to apply design to solve community problems. This study aims to explore the practical effectiveness of integrating USR into a design curriculum, thereby constructing a new model for USR fulfillment. Action research was applied in this study. The positive outcomes of integrating USR into the planning and implementation of a course titled 'Culture and Design Communication' were found to align with UNESCO's lifelong learning goals. The resulting model of USR fulfillment through design education will benefit the future sustainable development of university design education.

**Keywords:** university social responsibility; culture and design communication

## 1. Introduction

Considering global trends in education, the European Union put forward Project UNIBILITY in the Europe 2020 Strategy/Europe of Knowledge Initiative in 2015–2017, encouraging European universities to integrate social responsibility into major school development programs of the EU–University Social Responsibility (USR) Reference Framework. In 2017, the Ministry of Education Taiwan organized and promoted the Local University Social Responsibility Program, designating 'fulfilling social responsibility' as a key program for school development from 2018 onwards [1,2]. In addition to these overall development trends at home and abroad, the researcher also observed that design students lacked humanistic concern for the surrounding environment and the ability to adapt to social problems. Hence, in this study, the researcher attempted to understand and practice social responsibility by identifying a certain social problem and taking a design approach to problem-solving, thus helping students develop creative thinking and the ability to tackle social or environmental issues. With reference to previous research, this study posits that university design education has a responsibility to engage in social practices, starting by integrating such practices into formative education. Design educators and researchers should plan design curricula to accentuate university students' social responsibilities and narrow the gap between theory and practice. They should also integrate real social problems with the design discipline, adopt a problem-based approach, and develop design-based strategies to extricate the community from their plights, thereby realizing social practices.

### 1.1. Research Purposes

1. To integrate USR into 'Design Curricula' and induce university students' care for diverse cultures.
2. To explore the practical effectiveness of integrating USR into 'Design Curricula'.
3. To practice social responsibility through action research in design courses and to construct a model of sustainable USR fulfilment through 'Design Curricula'.

*1.2. Research Question*

1. Can integrating USR into 'Design Curricula' facilitate university students' care for diverse cultures and the community?
2. What are students' learning outcomes when USR is integrated into the course 'Design Curricula'?
3. What is the curriculum model involved in the integration of USR in 'Design Curricula'?

## 2. Literature Review

This section will elaborate on the development and analysis of university design curricula, as well as the significance of fulfilling social responsibility through design curricula. As design education falls under the umbrella of art education, studies related to social practices in art education are also discussed in the literature.

*2.1. Development and Analysis of University Design Curricula*

2.1.1. Development and Analysis of Problem-Based University Design Curricula

Historically, design curricula have been heavily weighted towards imparting professional knowledge and skill training, with a relative lack of connection with actual social problems. In recent years, many scholars have offered constructive analyses of art and design curricula in universities. Liao [3] stressed that art and design curricula in universities should unearth social problems by being integrated into the local community and considering the community as the field of the curriculum's practice. Design researchers and educators Hwang and Su [4] also mentioned that when the academic community departs from the classroom and confronts actual social scenarios, their problems will further extend into an effort to elucidate knowledge, and students will encounter the challenge of 'how to contribute to solving social problems'. When researchers and educators enter this field and develop a realistic understanding, they are faced with the exact challenges that must be addressed by scholars, cultural and social practitioners, and citizens. The above scholars' comments highlight the importance of social practices in art and design curricula. Jaehan [5] posited that art and design educators should not only reflect material and instrumental topics but also acknowledge the value of art practices in generating interpersonal interactions, connections, and relationships. This is because art practice is, in part, a social domain and platform that allows individuals to overcome boundaries and experience connections with other people. Although the importance of social practice has been illustrated by many design scholars, there is still a relative lack of connection between the theories and practices concerning which problems to address, which individuals to include to stimulate interactions, and which environments to consider. Hence, in addition to curriculum planning and teaching design principles, knowledge, and drawing techniques, design curricula should focus on how to cross existing boundaries and devise ways for students to interact and connect with educators, other students, or other parties from the perspectives of cultural innovation and social practice. Design curricula should also introduce environmental interventions into society in an attempt to solve real-life problems. By tackling actual social problems, design students will be challenged to propose design solutions, thus contemplating and integrating disciplinary practices and reviewing the prospects and sustainability of the learning outcomes of the design curricula.

2.1.2. Future Sustainable Prospects of University Design Curricula: Application of Creative Problem-Solving Model

According to Hsieh's [6] study, the creative problem solving (CPS) model can integrate design and creative processes more effectively than other creative pedagogical models and is applicable to design and creation. Previous research has shown that CPS-trained individuals are more creative than those who are not CPS-trained. In response to changing contexts, the former can produce solutions to problems and develop problem-solving abilities and creativity [6,7]. Thus, the CPS model is worthy of

use in design education. This model can be applied to design and creation through the following steps: (1) Find the rough initial directions by identifying a problematic direction from a group of possible directions, gathering data, and defining the target problems, (2) find solutions to the target problems based on design concepts, generate a large number of possible solutions, and form concrete ideas to develop new concepts corresponding to the problem context, (3) develop suitable visual representations of design concepts by selecting the target concepts and then apply semiotics to visualization, (4) finish the design work with the use of various visual principles and drawing techniques, and (5) evaluate the design from multiple aspects (creation, interview, and questionnaire). This five-step approach is adopted by educators during each stage of curriculum implementation.

*2.2. Significance of Fulfilling Social Responsibility through 'Design Curriculum'*

2.2.1. 'Design Curriculum' and Fulfilment of Social Responsibility

Brown [8] CEO of the world-renowned creative design company IDEO, expounded on the concept of design thinking by proposing that the design industry should overcome their preoccupation with objects and targets. Instead, he asserts that designers should more actively engage in overall social innovation, explore design opportunities pragmatically, and experiment in a human-centered and empathetic manner during the design process. This implies that design can be used as a strategy for solving social problems and, consequently, realizing social practice and innovation. Many educators have expressed their opinions on social practices in school curricula. Chen [9] stated that design is more than just a methodology. Rather, it is a perspective that necessitates actual interaction with society, the transformation of thoughts, and the breaking of traditions to turn 'design' into a force through 'social design'. Hwang and Su [4] also pointed out that social responsibility should be fulfilled by incorporating hands-on experience in traditional and abstract theoretical thinking in sociology. Actors should be reminded to focus on the actual current operations of society, reinterpret the social connections and needs of society, and look for viable solutions to connect theoretical knowledge with social practice and propose clear solutions to social problems.

In light of the positions of design scholars and researchers in Taiwan and abroad on the fulfillment of social responsibility in curricula, the present researcher intended to enable the content of design education to extend from the realm of professional design knowledge to discussions about the socio-environmental context. Design education should adopt a more open, diverse, and in-depth perspective to intervene in society and connect with the surrounding environment. In terms of the learning process, the fulfillment of social responsibility allows students to utilize knowledge representation and cultural reconstruction. In addition, the learning objectives stimulate students' future concerns for social problems and encourages their social participation.

2.2.2. Significance of Fulfilling Social Responsibility through the 'Culture and Design Communication' Course

In the past, courses on 'culture and design communication' focused mainly on imparting and explaining cultural theories and the principles of graphic design communication in a one-way lecturing form of teaching. These principles largely revolved around Pierce's semiotics [10] and Fiske's communication theory [11]. Cultural theories, on the other hand, were led by Hofstede's [12] cultural theory. The course analyzed how to obtain representative cultural elements from cultures, design based on different cultural contexts, and apply codes from the target culture to visual representation, thereby allowing students to engage in design communication using suitable cultural elements.

Previous research has proposed that art and design can be seen as a form of cultural creation and social reform. For instance, conceptual artist Kosuth [13] proposed the 'Artist as Anthropologist', suggesting that artists and designers tend to be more influential through their interpersonal connections during art interventions in a community. The power of art and design may bring about social change and awaken the power of a culture, thus guiding the community to transition from a reflection of local

history and culture to the development of a collective identity. Matarasso [14] pointed out that the power of art and design may engender social changes, as artists and designers enter into a community and lead the public in rediscovering aspects of daily life and reflecting on the surrounding culture and environment using their unique artistic and design models. Tung [15] also mentioned that art and design interventions in a community can motivate introspection towards the local culture, creating a collective identity and power for change. As design educators, the researchers kept these evolving connotations of art and design in mind when planning their teaching. Thus, this study attempted to transform 'design curriculum' into a strategic force in cultural creation and social reform and apply it to the curriculum planning and development of 'Culture and Design Communication'.

### 2.3. Summary: The Significance of Fulfilling Social Responsibility through the 'Culture and Design Communication' Course

Brown [8] stated that the value of design lies in the active intervention and argumentation involved in design thinking and problem-solving. Design relies on the interventions of humans, problems, and the environment to identify problems through interaction. The identified problems can be used to review one's own design concepts and look for innovative solutions. Only in this way can design transform into a strategy for solving social problems to achieve social practice and innovation. Yamazaki [16] indicated that 'design' is a problem-solving process and method. The design process brings individuals together and allows them to engage in collective planning and design centered around solving social problems, thus realizing social practice and innovation. Meanwhile, Keller and Sandlin [17] established art and design education as a form of collaborative engagement, during which creation is influenced by the extent of participants' involvement and interactions.

In consideration of the above research, the researchers in this study believe that design curricula can be extended and implemented through the introduction and in-depth exploration of community problems to cultivate learners' positive attitudes and subsequent respect towards diverse cultures, as well as their understanding of the community's culture. Design content should be rooted in problems encountered in the community and transformed and embodied in the form of design creation through learning and interacting with members of the local community. This study attempted to solve the various problems presented to the community by implementing a course, 'Culture and Design Communication'. The implementation of design curricula was employed as a method of cultural creation and social reform to guide students and tribal members from reflecting on the local history, culture, and industry of the community to developing a collective identity, allowing for the collaborative resolution of community problems and the fulfillment of USR.

## 3. Methods and Procedure

### 3.1. Research Participants

The target learners were sophomores from the Department of Creative Design at the university. This study attempted to integrate social responsibility practices into 'Culture and Design Communication', a year-two elective course offered in the second semester of 2018. A total of 42 students were enrolled in this course. The curriculum action research comprised three phases. Phase 1 was an introduction to the theories and principles in 'culture and design communication'. Phase 2 and Phase 3 were the social responsibility practicum I and II of the design course (please refer to Table 1 for the course structure). All the enrolled students possessed drawing and form-making skills, as well as 2D and 3D computer graphic techniques. They were divided into groups of five.

**Table 1.** Course Structure of 'Culture and Design Communication'.

| | Cycles | Course Content, Activities, and Teacher's Reflections | Problems Involved |
|---|---|---|---|
| **Phase 1** **Theories and Principle of 'Culture and Design Communication' and Insight into the Community Context** | **1st empirical cycle** <br> • Identify social problems <br> • Develop problem-related design concepts <br> • Produce concept-related visual representations <br> • Finish visual design <br> • Hold a student work exhibition <br> • Evaluate design via multiple assessment methods <br> • Lay the groundwork for the next action | Week 1: Introduction to cultural theories, the Pnguu culture, and history of the Tsou tribe <br> Week 2: Introduction to principles of communication, the semantics of semiotics, and case study <br> Week 3: Field trip to the Pnguu community to understand the community's resources, context, and needs <br> Week 4: Visual design using Pnguu cultural codes, communication principles, and drawing techniques <br> Week 5: Practical design and production <br> Week 6: Phase 1 student work exhibition <br><br> **Teacher's reflections** <br><br> 1. The students only had a surface-level understanding of the community and produced visual designs that were largely decorative without any in-depth experience with the local culture. <br> The students pointed out the need to understand the community culture in greater depth. The Tsou tribe hoped the products would carry more of their cultural essence. <br> 2. Teaching activities in Phase 2 were designed based on these reflections. | 1. The Pnguu community was in need of a visual image <br><br> 2. The Pnguu community needed a visual image to create community cohesion |
| **Phase 2** **Social Responsibility Practicum I in 'Culture and Design Communication'** | **2nd empirical cycle** <br> • Identify social problems <br> • Develop problem-related design concepts <br> • Produce concept-related visual representations <br> • Finish the designs <br> • Hold a student work exhibition <br> • Evaluate designs via multiple assessment methods <br> • Lay the groundwork for the next action | Week 7: Field study at the Pnguu community and in-depth interviews with community experts, community builders, and chairmen to understand their needs and problems. <br> Week 8: Bamboo weaving and woodcarving workshops co-organized with the community for students to learn from community experts. <br> Week 9: Applying cultural theories to communication principles and case study. <br> Week 10: Practical design of cultural and creative products using community-taught principles and skills of bamboo weaving and woodcarving. <br> Week 11: Practical design and production of cultural and creative products <br> Week 12: Phase 2 student work exhibition. <br><br> **Teacher's reflections** <br><br> 1. The students found it interesting and fulfilling to integrate design into bamboo weaving and woodcarving <br> The tribal members generally found the students' work creative and culturally relevant <br> 2. The tribal members identified artefact preservation as a more urgent task at hand and the need for the re-planning of their craft museum <br> 3. Teaching activities in Phase 3 were designed based on these reflections | 1. The tribal members were gradually forgoing their traditionally significant ways of living—bamboo weaving and woodcarving <br><br> 2. The tribal members expressed a need for packaging design for cultural and creative products |
| **Phase 3 Social Responsibility Practicum II in 'Culture and Design Communication'** | **3rd empirical cycle** <br> • Identify social problems <br> • Develop problem-related design concepts <br> • Produce concept-related visual representations <br> • Finish the designs <br> • Hold a student work exhibition <br> • Evaluate designs via multiple assessment methods <br> • Lay the groundwork for the next action | Week 13: Introduction to design principles in spatial exhibition and space planning Week 14: Field study at local craft houses run by tribal members to understand their problems Week 15: Case study in space planning and exhibition design Week 16: Practical design in the overall planning of local-run craft galleries in exhibits, taxonomy, and exhibition design Week 17: Practical design through transforming individually-run craft galleries into street corner museums Week 18: Phase 3 student work exhibition <br><br> **Teacher's reflections** <br><br> 1. The tribal members were fond of the students' designs, and the students gained a sense of accomplishment <br> 2. A sustained effort to explore new community problems as the basis for planning new course actions in the next semester gave impetus to continuous improvement in design | 1. There was a need to preserve the Tsou artefacts in the community <br><br> 2. The tribal members thought that their individually-run craft galleries were decrepit, chaotic, and in need of some design and planning. They suggested remodelling them into street corner museums to improve their images |

*3.2. Teaching Field*

The Pnguu community of the indigenous Tsou tribe in the Alishan Township, Chiayi County, Taiwan was designated as the site of teaching. According to a study by Hwang and Su [4], the Pnguu tribe is located in the Pnguu Village in Alishan Township along the Alishan River at the foot of Tashan, the sacred mountain of the Tsou tribe. The tribal members gain their livelihood by planting beans, fruits, and vegetables and hunting. The Pnguu community was selected as the study area because the tribal members cared about their own culture but had no way to revitalize it. In addition, the researcher had taken part in the Ministry of Education's USR program in Taiwan since 2018. The local Pnguu Village elders and chairman of the Pnguu community association have talked to the researcher to offer help in continuing their tribal culture, and they need design to develop their culturally local industry in their community.

*3.3. Research Procedure*

### 3.3.1. Diagnose and Define the Problem

In the past, this design course was confined to lectures and explanations of cultural theories through visual presentations of semiotic and design communication principles. The researchers observed in the teaching environment that, despite the students' proficiency in creating aesthetic and communicative visual artwork, they displayed a lack of true understanding and cultural concern for the social environment in which they grew up. Hence, an effort was made to adjust the course and allow them to apply the theories they learned to social problems.

### 3.3.2. Devise a Feasible Action Plan

(1) Formulate the course structure and content: This course lasted for 18 weeks in total. Over the span of six weeks, the students were required to visit the Pnguu community and identify existing problems before applying the CPS model to propose detailed solutions. The CPS model was applied through the following procedure. Step 1: Identify social problems. Step 2: Develop problem-related design concepts. Step 3: Produce concept-related visual representations. Step 4: Finish visual representation design. Step 5: Hold a student work exhibition. Step 6: Evaluate design via multiple assessment methods. Step 7: Lay the groundwork for the next action. The implementation framework and content of this course are shown in Table 1.
(2) Designate the Pnguu community as the venue of teaching.
(3) Seek help from the Community Development Association and community experts.

### 3.3.3. Reflect and Re-Plan

A spiral model for the action research cycles was adopted, in which reflection on the pedagogical research in Phase 1 was used to plan the teaching practices in Phase 2, and reflection on the teaching practices in Phase 2 was used to plan the same in Phase 3. The three phases of the action research were thus interconnected.

### 3.3.4. Take Action and Incorporate CPS

This course intended to incorporate six weeks of practical fieldwork in the Pnguu community, during which the student groups identified community problems and developed design concepts in response to the target problems. They then embodied the selected concepts via visual representation and organized a student work exhibition after finishing the design products. Their designs were evaluated using multiple assessment methods, and the results laid the groundwork for their next action. The actions taken are shown in Table 2.

**Table 2.** Identified problems and details of actions in each phase of the action research.

| Fields of Teaching | | Identified Problems | Implications of Actions |
|---|---|---|---|
| Phase 1<br>The Pnguu community | 1.<br><br>2. | The tribal elders were worried about cultural decay.<br>The community expressed their need for our help in developing a visual identity to create community cohesion. | Visit the community to understand their resources and representative codes, including the people, culture, land, scenery, production, rituals, and tales. Construct the visual image of the community using the collected codes. |
| Phase 2<br>Pnguu Lishan SchoolCraft houses in the Pnguu community | 1.<br><br>2.<br><br><br><br>3. | The Tsou tribe hoped to see more of their cultural essence in the products.<br>The elders worried that traditionally significant crafts such as bamboo weaving and woodcarving would become obsolete<br>The tribal members expressed the need for cultural and creative products for marketing purposes. | Attend community-held workshops to learn traditional bamboo weaving and woodcarving skills and apply them to design practices. |
| Phase 3<br>Craft houses in the Pnguu communityExhibition Hall 3 of the NYUST | 1.<br><br><br><br>2. | The individually run craft galleries had no aesthetic design. The exhibits were uncategorized, and the exhibition space had a chaotic circulation flow.<br>The tribal members hoped to re-organize, re-plan, and re-model the individually run craft houses into the street corner museums of Pnguu. | Utilize design to tackle the identified problems regarding the local craft galleries.<br>Facilitate the re-organization and re-planning of the local craft galleries to promote cultural sustainability. |

### 3.3.5. Apply Research Tools

The tools included interviews with student representatives, a questionnaire survey, photographic records of classes, the teaching assistant's observation log of class activities, assessments of students' creations, a teaching diary, and interviews with tribal members.

### 3.3.6. Disseminate Knowledge and Hold Exhibitions

Open exhibitions were held in Week 6, Week 12, and Week 18, respectively, in which the tribal members, local elders, teacher, and students interacted. The design works developed by the students were available to the tribal members at no cost. The action research in this course formed a dynamic cycle of reflection and action, as shown in Table 1.

## 4. Result Analysis

The coding of the data collected during course implementation is illustrated in Table 3. The results of the pre-course questionnaire survey and end-of-semester course evaluation are shown in Tables 4 and 5, respectively. For each of the three phases of course implementation, photographic records of the field studies and the students' design products are displayed in Table 6 and Figures 1–3, respectively. The interview questions for students and their tribal members are outlined in Appendices A and B.

**Table 3.** Coding of the collected data.

| Research Tool | Example | Description |
|---|---|---|
| Interviews with students | S1-I-Q1-1021 | S means student. 1 means the first student. I means interview. Q means interview question. 1021 means date. |
| Reflective teaching diary | TD-1021 | TD means teaching diary. 1021 means date. |
| Interviews with tribal members | A1-I-Q1-0107 | A1 means tribal member. I means interview. Q means interview question. 0107 means date. |
| Teaching assistant's weekly observation log | TA-Week01-01 | TA means teaching assistant. Week01 means number of weeks. 01 means document code. |
| Assessment of students' creations | SC-Week06 | SC means student's creation. Week06 means number of weeks. |

**Table 4.** Results of the pre-course questionnaire survey.

| Questionnaire Item | Mean |
|---|---|
| 1. You are interested in indigenous cultures | 2.92 |
| 2. You concern yourself with indigenous cultures | 2.81 |
| 3. You are fond of indigenous cultures | 2.91 |

Note: Items were scored on a five-point scale. A higher score represented a higher level of agreement.

**Table 5.** Results of End-of-Semester Course Evaluation.

| Questionnaire Item | Mean |
|---|---|
| 1. This course allowed you to step into the Pnguu community to identify its problems | 4.68 |
| 2. This course allowed you to help the Pnguu community solve their problems with your design specialty | 4.64 |
| 3. Learning in the Pnguu community allowed you to put your professional design knowledge into actual practice | 4.55 |
| 4. Learning in the Pnguu community allowed you to gain more disciplinary knowledge | 4.65 |
| 5. A design course centered on solving the Pnguu's community problems could improve your ability to integrate practice and course content | 4.86 |
| 6. A design course centered on solving the Pnguu's community problems could improve your knowledge competency | 4.58 |
| 7. A design course centered on solving the Pnguu's community problems taught you to care for different cultures | 4.65 |

Note: Items were scored on a five-point scale. A higher score represented a higher level of agreement.

**Table 6.** Photographic Records of Class Activities in Each of the Three Phases.

---

**Phase 1 Field: The Pnguu Community**

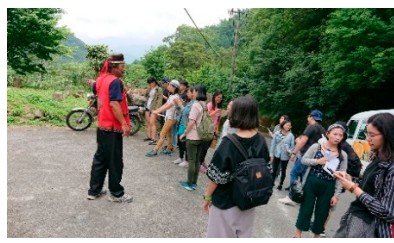

The village chief guiding students through the Pnguu community. (TA-Week01-04)

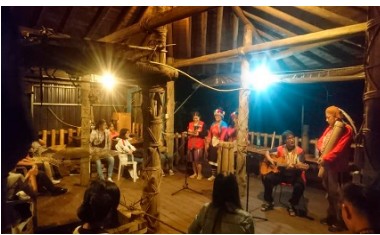

Indigenous members singing a Tsou folk song. (TA-Week01-05)

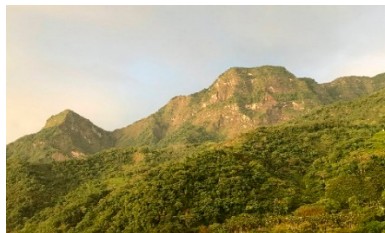

Tashan, a spiritual symbol of the Pnguu community of the Tsou tribe. (TA-Week01-06)

---

**Phase 2 Field: Bamboo Weaving and Woodcarving Workshops in the Pnguu Community**

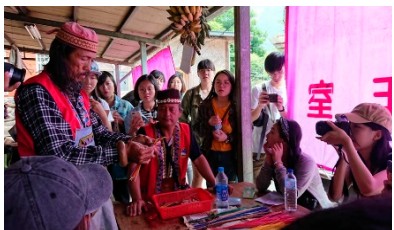

A bamboo weaving expert teaching students how to make bamboo strips. (TA-Week07-03)

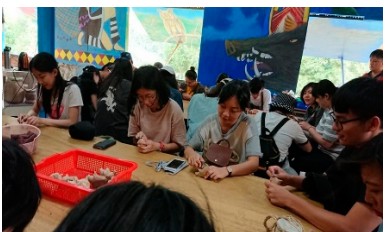

A woodcarving expert teaching students how to make boar-themed crafts. (TA-Week07-04)

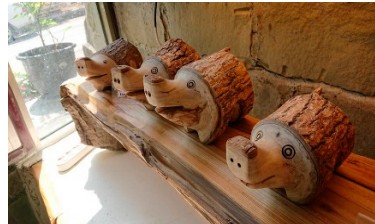

Works by woodcarving expert Chen. (TA-Week08-03)

---

**Phase 3 Field: Craft Houses in the Pnguu Community—Laus, Lanhou, and Moar Craft Houses**

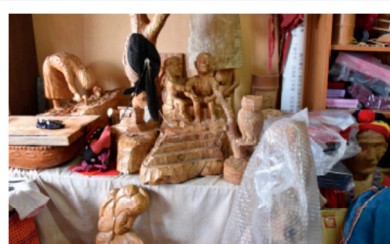

Laus Craft House (TA-Week13-02)

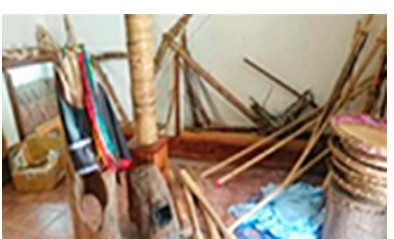

Lanhou Craft House (TA-Week13-04)

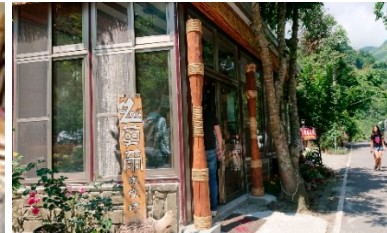

Moar Craft House (TA-Week13-05)

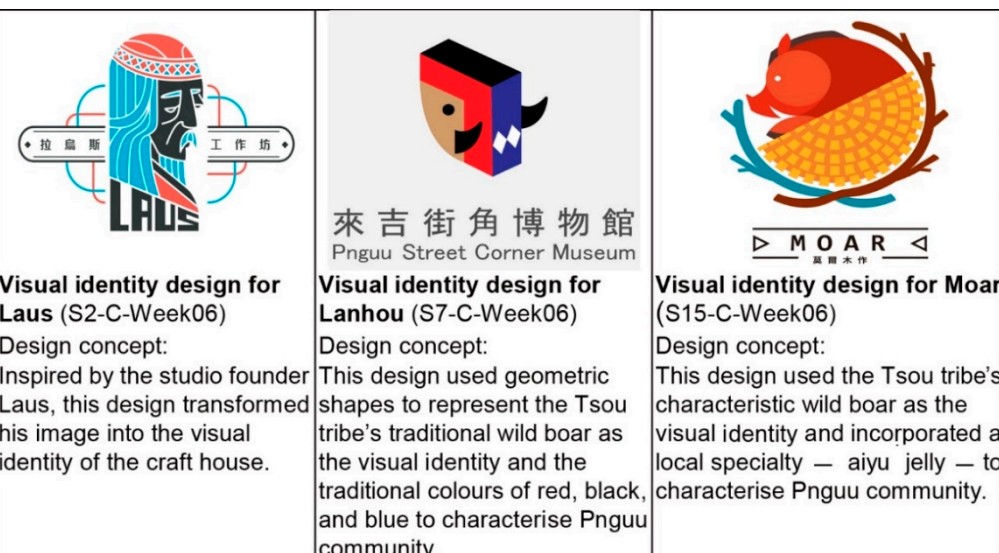

**Figure 1.** Students' design creations in Phase 1.

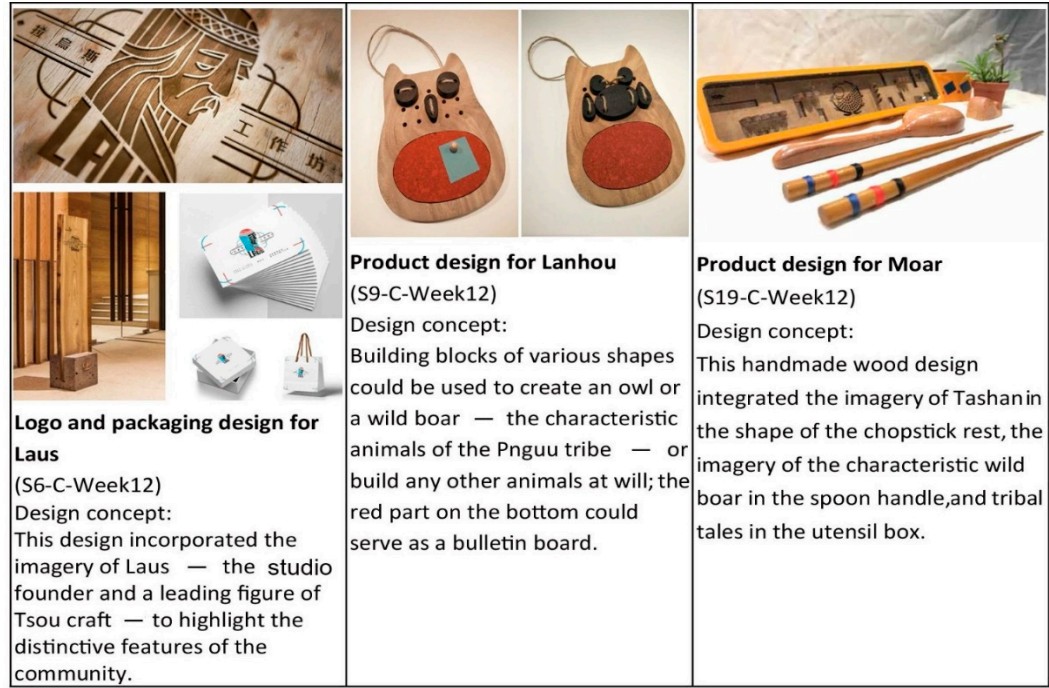

**Figure 2.** Students' design creations in Phase 2.

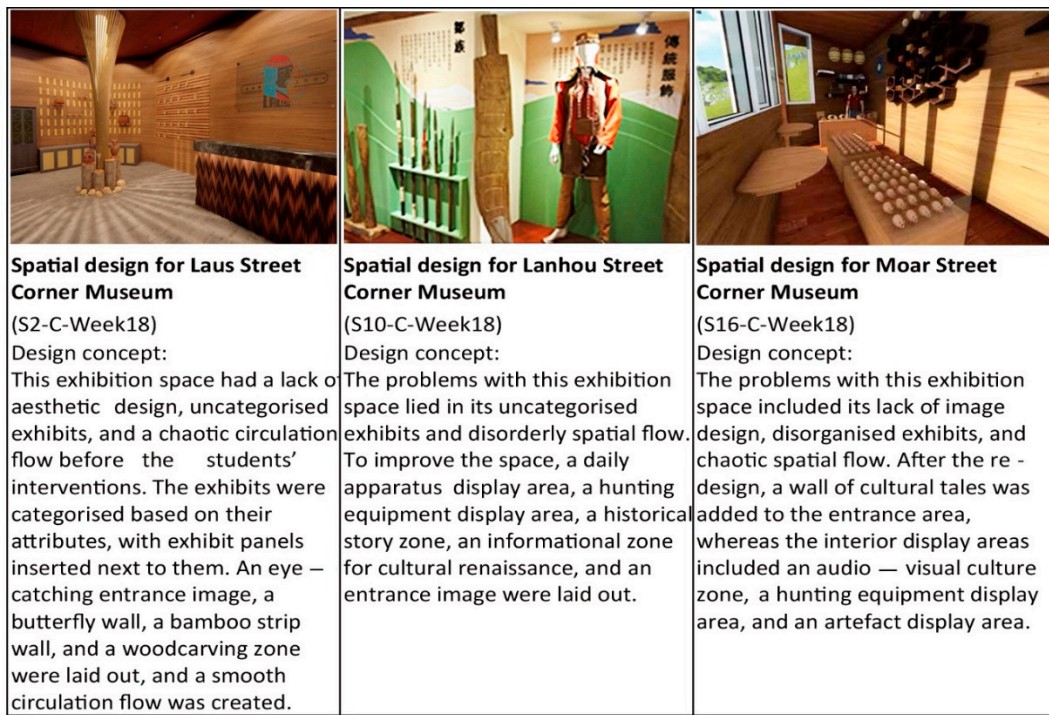

| Spatial design for Laus Street Corner Museum | Spatial design for Lanhou Street Corner Museum | Spatial design for Moar Street Corner Museum |
|---|---|---|
| (S2-C-Week18) | (S10-C-Week18) | (S16-C-Week18) |
| Design concept: | Design concept: | Design concept: |
| This exhibition space had a lack of aesthetic design, uncategorised exhibits, and a chaotic circulation flow before the students' interventions. The exhibits were categorised based on their attributes, with exhibit panels inserted next to them. An eye — catching entrance image, a butterfly wall, a bamboo strip wall, and a woodcarving zone were laid out, and a smooth circulation flow was created. | The problems with this exhibition space lied in its uncategorised exhibits and disorderly spatial flow. To improve the space, a daily apparatus display area, a hunting equipment display area, a historical story zone, an informational zone for cultural renaissance, and an entrance image were laid out. | The problems with this exhibition space included its lack of image design, disorganised exhibits, and chaotic spatial flow. After the re - design, a wall of cultural tales was added to the entrance area, whereas the interior display areas included an audio — visual culture zone, a hunting equipment display area, and an artefact display area. |

**Figure 3.** Students' design creations in Phase 3.

*4.1. Implications of Course Implementation*

4.1.1. Using the Fulfilment of Social Responsibility AS the Underlying Ideology and 'Community Problems' As the Course Focus Turned Students from Theory Learners into Individuals Who Cared about Diverse Cultures and Acted to Solve Social Problems

- An outcome of course implementation was to nurture students' care for different cultural groups. A pre-course questionnaire survey (Table 4) was conducted in Week 1. The results attested to a lack of concern for Taiwan's indigenous cultures among the students. Nevertheless, the end-of-semester course evaluation (Table 5) yielded different results and showed that the students had learned to care about different cultural groups (the mean scores in items 4–7 exceeded 4.5, as shown in Table 5).

- Students acknowledged diverse ethnic cultures and became aware of their own social responsibilities. This is evidenced by the following excerpts from the end-of-semester interviews with the students:

  "The Tsou culture is a gem. We should try to do something for the Pnguu community and sustain their culture."

  (S7-I-Q1-0119)

  "I realized my social responsibility and believed that I could give back to society by helping the Pnguu community with my own skills."

  (S4-I-Q1-0115)

- A thorough understanding of 'community resources' gave students an insight into the Tsou tribe's culture and added depth to their design. This is supported by excerpts from the students from the interviews in Phase 2:

  "Using local resources in my production gave me a better understanding of the community characteristics and inspired more design ideas."

  (S08-I-Q1-1203)

"Taking resources from the local community allowed for the formation of the design's unique cultural characteristics."

(S15-I-Q1-1203)

### 4.1.2. With the Philosophy of Fulfilling Social Responsibility and a Focus on 'Community Problems', the Design Course Gave Tribal Members a Sense of Pride

The following excerpts from the interviews with tribal members could attest to this:

"You helped us a lot. These designs made us feel proud and special."

(A1-I-Q5-0118)

"By stepping into Pnguu and helping the community, the students established a mutually beneficial tie between the school and the community."

(A2-I-Q1-0119)

"Introducing students to the uniqueness of the community could help the community and render more external resources accessible."

(A3-I-Q2-0121)

### 4.1.3. With the Philosophy of Fulfilling Social Responsibility and a Focus on 'Community Problems', the Design Course Contributed to Solving Social Problems and Gave Rise to a Sense of Accomplishment in Learning

This was substantiated by the following excerpts from the student interviews:

"It felt incredible and fulfilling to have my design seen by the tribe and to return our products for them to use."

(S2-I-Q2-0115)

"To know that my design was actually used, exhibited, and seen by the tribe, it gave me a sense of accomplishment."

(S14-I-Q2-0120)

### 4.1.4. Integrating 'Social Responsibility Practice' into the Planning of 'Design Curriculum' Entailed Core Aspects of Thinking, Including Problem Introduction and Reflection, Intervention into Fields, and Intervention and Interaction of Roles

- Design theories as foundational: Design students must acquire a solid foundation in design knowledge and theory. Only with sufficient knowledge can they solve social problems and validate the theories they have learned in real-life scenarios.
- Intervention into fields. It is important to emphasize participation and cross-domain integration, including the diverse choices of locality and the cross-domain nature of collaborative teaching, which are key factors that should be taken into consideration during curriculum planning. Different fields involve different problems and resources, which stimulate students to have a diverse mindset.
- Problem introduction and planning: This study found that teachers can 'propose' problems repeatedly based on events occurring in students' surroundings to improve the curriculum and drive conversations. Through the 'proposal of problems', curriculum planning connects with social, cultural, environmental, and industrial elements, thus elevating the value of the design curriculum.
- Intervention of roles: Integrating 'social responsibility practices' into the design curriculum constructed a co-learning and interactive context in which a community was formed between

students and teachers, community experts, and other students and among tribal members, thus generating an interactive force for mutual learning. The following section details each of these roles.

Students: Through solving community problems, students learned how to communicate with the tribal members and how to apply their knowledge and skills to problem-solving. This was completely outside the realm of static classroom learning. By interacting with the community, the students were able to confront the everyday struggles of the community and attempt to solve real-life problems using their design knowledge and skills. This process enabled them to learn, reflect, and better recognize the gaps in their knowledge, thereby motivating them to engage in active learning. In addition to the abstract theories taught at the university, this course focused on teaching students how to apply such academic theories to actual social problems. During this process, students learned how to put academic theories to use and gained hands-on experiences through problem-solving. Honing their problem-solving skills in a real-life community context and tackling problems by leveraging their university knowledge and seeking cooperation with experts from other fields allowed students to develop a concern for different ethnic cultures and interest in social–public affairs.

Teachers: This study found that, to break with tradition, teachers must have a reflective and adventurous attitude. The practice of guiding students into the community adds much more diversity to teachers' early course preparation and planning. It also brings them more entertaining teaching fields and content, thus boosting students' interest in their education. This leverages the diverse roles played by teachers and is beneficial for local development and industrial transformation.

Tribal members: Through a constant effort to understand tribal members' needs and by gathering feedback from tribal members, this study developed a major impetus to sustain the community fieldwork. The interactive discussions enabled both teachers and students to respond to local community issues more sensitively. During the post-course interviews, the tribal members opined that the students truly helped the community and opened their eyes to the many other possibilities for artifact preservation. They also began to take greater pride in their community.

- Interaction of roles. According to this study, shifting the learning style to a practice-led process of co-learning and interaction enabled each practical stage to engage different social groups, as well as students and community experts with different specialties. This resulted in more diverse and frequent interactions and created synergistic benefits. It also allowed students to exchange ideas with people from other fields, offsetting each other's weak points and allowing them to learn more from the design process. This is supported by the following interview excerpts:

"It helped to hone our disciplinary skills (graphic design, product design, and spatial design)" integration of design and allowed for more systematic reflection on different needs during the design process."

(S1-I-Q4-0116)

"Discussing the design with team members with different specialties, summarizing different opinions, and learning different knowledge, it was incredible."

(S12-I-Q4-0118)

*4.2. Course Evaluation*

After the three phases of course implementation, the design course was reviewed based on interviews with the students and tribal members.

4.2.1. The Execution Time Was Too Short

"Students thought that the visits to Pnguu community were too brief."

(S4-I-Q5-0116)

"Our work was not elaborate enough because the design execution was too hurried. We might have been able to contribute more to the tribe if we had enough time."

(S6-I-Q5-0117)

### 4.2.2. More Persistent and In-Depth Communication with Tribal Members Was Warranted during Design Conceptualization and Production

"More in-depth discussions are required during design conceptualization, only then can the product satisfy the tribe's needs."

(S10-I-Q5-0117)

## 5. Discussion

In 2018, the Ministry of Education in Taiwan officially launched the Higher Education Cultivation Project, with fulfilling social responsibility as one of its four major goals [2]. The purpose of the USR projects is to align universities with the future development directions of higher education. The core values of the USR projects are local connections and talent cultivation, which provide universities with a people-orientated approach and a focus on local requirements to meet their social responsibilities by assisting in solving regional problems with humanistic concerns [2]. Teachers and students from different departments, disciplines, and universities are encouraged to contribute their abilities and knowledge to industrial and cultural development and thus help extract material benefits for regional development that are tangible for local industries. To align with the purpose of the USR Projects, the Pnguu community of the indigenous Tsou tribe was designated as the field for the active involvement of students, teachers, and indigenous residents. USR needs the active involvement of students in solving a "requirement" identified in the community, the clarification of techniques, or the knowledge to be developed, and the provision of spaces intentionally organized for reflecting upon the experience [18]. By empowering Penguu indigenous residents to develop their industries with knowledge of design, local art craft, and regional ecology materials, the authors used design curriculum as a strategy to get students involved in these coordinated projects, which, in turn, advanced their practical know-how, ways to communicate with people, and ways to care for people. Design curricula are first used as a strategy for helping remote indigenous communities.

A design course for 'Culture and Design Communication' was applied to get students actively involved in this research. Based on community culture as the developmental spindle, a field for a practical creation exercise was provided to encourage learners to present doubts about their assumptions and absorb more knowledge and practical experience to help them re-think the role played by the Penguu indigenous community. This allowed students to listen to the feedback of the indigenous people with different cultures and encouraged students to further understand indigenous Tsou culture and induce multicultural interest. The course content and objectives of 'Culture and Design Communication' aim to determine the problems, behavior modes, and communication methods of people from regions with distinct cultures, as well as unique codes, beliefs, norms, and values [12]. Design is a medium between culture and industry as well as a strategy. Learners have to understand distinct cultures and respect and comprehend different cultural traits, dilemmas, and demands to express the design concept [6]. Using the practice of social responsibility through the design curriculum, the indigenous cultural trait problems, behavior modes, and communication methods of the Tsou Penguu Community were discovered to assist in the sustainable development of the indigenous cultural industry and solve community problems so that students could understand indigenous culture and further respect it. The discussion in these two paragraphs forms the first portion of the conclusion.

The university's social responsibility was fulfilled through the design course of 'Culture and Design Communication' in this study, and, according to the above-mentioned course content and objectives, six core curriculum development dimensions are concluded. 'Problem introduction and planning', 'intervention into fields', and 'intervention of roles' correspond to the design concept of Brown [8], the executive of IDEO, a famous professional design company, showing that the value of

design exists in the positive intervention and dialectic of design thinking and problem-solving. Design needs the intervention of people, problems, and environments to discover the existence of problems through interaction. Design, with the inspection of personal design ideas through problems and the innovative discovery of 'solutions', could be used as a strategy to solve social problems and achieve social practice and innovation. These three dimensions also correspond to Yamazak [16], who showed that 'design' is a process and method used to solve problems, where people are connected in the design process, and making plans and using design to solve social (community) problems are the spindle for achieving the value of social practice and social innovation. Aligning with Costandius' research [19], art (design) is, in fact, an effective medium to enhance social responsibility.

According to the practical experience in Section 4.1.4 and the suggestions in Section 4.2, 'design theories as basis', 'interaction of roles', and 'construction of a learning community' were added. It was discovered during the course practice that it is difficult for learners without professional knowledge of design to completely deal with and solve problems in real situations. It is, therefore, necessary to add the dimension of 'based on design professional theory'. Furthermore, 'interaction of roles' and 'construction of a learning community' were ignored in previous studies [3,4,8,16]. During the course practice in this study, it was discovered that the learning style should be changed into a practice of common interactive learning, allowing different communities, such as students, community enterprise masters, and community organizations, to participate in each practice process and allow the interaction among communities to become more diversified and frequent to construct a learning and interactive situation among students, between teachers and students, between students and community masters, and among indigenous people to form a commonwealth and generate interactive energy for mutual learning. The interactive channels contain networks and technological social media to develop these synergistic benefits. In this case, 'interaction of roles' and 'construction of a learning community' were added. The learning outcomes of the roles in the interactive process are explained as follows.

(1) Instructors guiding students in the community offer a more vivid and interesting teaching field and greater quality to enhance students' learning interests, thereby developing the power of teachers' diverse roles, assisting in local development and problem-solving, and producing positive effects on the society.

(2) Students learn to communicate and discuss with indigenous people as well as apply knowledge theories and skills in the community problem-solving process, which is completely different from static learning in classrooms. This allows students to learn to apply abstract academic knowledge, acquire practical learning in the problem-solving process, and implement university social responsibility in the course practice process to put what they learn to good use. This process affords students a sense of accomplishment and reality and could cultivate students' concern about different ethnic cultures and help them pay attention to the public agenda in society and formulate a basis for the sustainable development of university design education.

(3) Based on the after-course interview, indigenous people considered that the connections between students and communities could actually help the communities. This helped the indigenous people realize more possibilities to preserve their culture and acquire new ways of thinking. This process further influenced the indigenous people's willing to pass down Tsou culture with different methods, gradually presenting a sense of honor about Tsou culture and reflecting the uniqueness and development methods of Tsou culture. It also opened a new possibility for cultural revitalization and indigenous culturally sustainable development.

During the research process, the introduction and interaction of 'people', 'fields', and 'problems' with constant spiral cycles is the core power of design to achieve social innovation. Instructors could constantly "propose" problems as the driving force for course improvement and spiral dialogue, connecting the "fields" of society, culture, environment, ecology, and industry with "persons", including teachers, students, the community's indigenous people, and industry experts through the course

planning of "proposing problems", thus enhancing the value and content of the design course. The discussion in this paragraph shapes the third portion of the conclusion.

## 6. Conclusions and Recommendations

The analysis of the course outcomes indicates that integrating USR fulfillment with design curricula and using social problems as the basis of curriculum development can offer students a way to engage in hands-on design practices. When a design curriculum is transformed into a means of cultural creation and social reform, it drives students to engage with indigenous discussions and reflect on local history, culture, and industry through collective problem-solving, thereby engendering social improvement and cultural innovation and indigenous culturally sustainable development. This aligns with UNESCO's core initiatives in lifelong learning: Learning to be, learning to know, learning to do, learning to live together, and learning to change. Based on the discussion in Section 5, this study drew the following conclusions:

(1) Using the course 'Culture and Design Communication' to fulfill USR allowed students to experience social responsibility and develop care for diverse cultures. Their contribution to solving community problems gave them a sense of accomplishment and gave the tribal members a sense of pride.

(2) The curriculum development model of fulfilling university 'social responsibility' through 'Culture and Design Communication' entails six core aspects: 'Design theories as basis', 'problem introduction and planning', 'intervention into fields', 'intervention of roles', 'interaction of roles', and 'construction of a learning community' and forms the basis for the sustainable development of university design education.

(3) A deeper understanding of community problems motivates teachers to sustain their efforts in bringing students into the community to solve problems. Hence, a spiral cycle of the repeated introduction and interaction of 'people', 'fields', and 'problems' is the key engine driving design towards social innovation and culturally sustainable development.

This study proposes the following recommendations:

(1) It is necessary to introduce fieldwork sites and undertake long-term and in-depth field investigations. The interviews with students showed that gaining a deeper understanding of the Pnguu community required more time and in-depth discussions. To delve further into the problems and successfully extend help to the community, it is necessary to conduct long-term explorations in this field.

(2) Online learning communities for 'students', 'teachers', 'community experts', and 'workshops' should be constructed to facilitate after-class interactions and communication. This study found that building a learning community is more beneficial than independent learning. Feedback from the course evaluations indicates that a learning community should be established to reinforce the 'interaction of roles' and 'cross-domain' integration and promote after-class communication and discussion between students and tribal members to craft know-how and skills. The learning community may also take advantage of social media (such as Facebook and Line).

(3) Design curricula are first to be used as a strategy for helping remote indigenous communities. They can then be expanded to areas with other cultures or adopted to other strategies in the future.

**Funding:** This research received no external funding.

**Acknowledgments:** I am grateful to Ministry of Education Taiwan and all study participants for their contributions.

**Conflicts of Interest:** The author declares no conflict of interest.

## Appendix A

Interview Questions for Students

1.    After Phase 1

    (1)    Do you have any suggestions for this part of the course?

    (2)    What are your thoughts on the student work exhibition in the Pnguu community?

    (3)    What more support do you need to gain better insights into Pnguu culture?

2.    After Phase 2

    (1)    Which part of the bamboo weaving and woodcarving workshop impressed you the most?

    (2)    Are you influenced by other students' designs and ideas? Which are your favorite ideas?

    (3)    Do you find it interesting to integrate bamboo weaving and wood carving techniques into the design of modern lifestyle items?

    (4)    Does the Pnguu community have any other problems to be solved?

3.    At the End of the Semester

    (1)    Does this course make you aware of your own social responsibility?

    (2)    Do you think designing with community resources better demonstrates the outcomes of social responsibility practices?

    (3)    Did this course give you insight into how to help the local community with your expertise?

    (4)    Did selecting the Pnguu community as the site of fieldwork allow you to gain more cross-domain knowledge?

    (5)    Do you have any other advice or thoughts on this course?

**Appendix B**

Interview Questions for Tribal Members

(1)    Are the students' design products effective in solving community problems?

(2)    Do you feel more motivated to revitalize the Pnguu culture with the help of students?

(3)    Does interacting with the students give you better insights into the value of your culture or more ideas about promoting your culture?

(4)    Do you like the students' design? Do they have any room for improvement?

(5)    How do you feel after seeing the students' design?

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
