# Peer review of "Applying Action Research in Design Curricula to Fulfill University Social Responsibility—A Case Study of the Pnguu Community of the Tsou Tribe"

_sustainability, doi:10.3390/su11247132_

Round 1

Reviewer 1 Report

I find the project as conceived and executed to be thoughtful, worthwhile and from my point of view of considerable value to both sets of participants.  I am very impressed by the photographic exhibits and the level of sophistication in blending both sides of the cultural divide, as I am with the redesign of the exhibit halls.

My sense is that for the students the overall long term value lies in getting out of a classroom setting and being fully engaged in another cultural setting into which ultimately they are invited.

There is minimal “clean up” of minor grammar and usage errors but overall the presentation is quite

Author Response

Point1 : I am very impressed by the photographic exhibits and the level of sophistication in blending both sides of the cultural divide, as I am with the redesign of the exhibit halls. There is minimal “clean up” of minor grammar and usage errors but overall the presentation is quite...

Response1: I use the "Track Changes" function in Microsoft Word to modify the full text.

Reviewer 2 Report

Social responsibility is an issue discussed nowadays not only as a voluntary activity; and a general University Social Responsibility model has been declared. Thus, it deserves attention and looking for different effective models is a natural process.

The presented article clearly describes the model of USR integration to university education process where creative problem solving was applied in action research realised with Pnguu community. The authors used triangulation of methods to increase the validity and reliability of the results.

Three remarks directly to the text connected:

ll180-182 probably should be omitted. ll-212-213 "the students were required to visit the Pnguu community and identify existing problems before applying the CPS model to propose detailed solutions" - Can the author be more specific? What was the focus / Was the focus specified? The author may consider adding words/phrases to key words as e.g. creative problem solving.

What I miss in the text is the sustainability aspect;the title suggests that the author deals with the sustainability development and there is no connection to sustainability presented in the article. There have been studies published on Universities' Social Responsibility and Sustainable Development that the author may introduce. This should be added to the introductory part of the text and then the author should relate their results to the studies/research already conducted.

Author Response

Point1: Line180-182 probably should be omitted.

Response 1: Line180-182 have been omitted. Please view line179 in the revised submission.

Point2: line-212-213 "the students were required to visit the Pnguu community and identify existing problems before applying the CPS model to propose detailed solutions" - Can the author be more specific? What was the focus / Was the focus specified? The author may consider adding words/phrases to key words as e.g. creative problem solving.

Response 2: The specific steps of CPS has been added, please view line 209-213.

Point3: What I miss in the text is the sustainability aspect; the title suggests that the author deals with the sustainability development .....

Response 3: The title has been revised as “Applying Action Research in Design Curricula to Fulfil University Social Responsibility—Working with the Pnguu Community of the Tsou Cultue.” Please view line 2-4 in the first page.

Reviewer 3 Report

Title: Design Intervention into the Community to facilitate sustainable development of university education: Fulfil University Social Responsibility through Design Curriculum

Study that addresses university social responsibility from a practical perspective. The objective of the study, the design and applied methodology is correct. However, the title, at first, does not make it easy to know the main theme of the study and its objective. When dealing with social practices in art education, it would be convenient if something appears in the title, since the University Social Responsibility does not work the same in all areas of knowledge.

However, the article addresses university social responsibility from the transfer of knowledge to disadvantaged areas. This is a type of work by transversal competences by university students, but it is not the only one.

It is convenient that the authors contextualize in greater depth what the university social responsibility consists from a broader perspective, to then be able to focus the objective of the study. Example: line 387-388: The curriculum development model of fulfilling university ‘social responsibility’ through ‘Culture and Design Communication’ entails six core aspects:

It would be convenient to include a discussion section, where the results achieved could be compared with those of other studies.

Author Response

Point 1: The title, at first, does not make it easy to know the main theme of the study and its objective. When dealing with social practices in art education, it would be convenient if something appears in the title.

Response 1: The title has been revised as “Applying Action Research in Design Curricula to Fulfil University Social Responsibility—A Case Study of the Pnguu Community of the Tsou Culture.” Please view line 2-4 in the first page.

Point 2: It is convenient that the authors contextualize in greater depth what the university social responsibility consists from a broader perspective, to then be able to focus the objective of the study.

Response 2: What the USR consists from a broader perspective is contexturised in discussion section, please review line 372-423.More related research are reviewed and presented in section 5.

Point3: The curriculum development model of fulfilling university ‘social responsibility’ through ‘Culture and Design Communication’ entails six core aspects: It would be convenient to include a discussion section, where the results achieved could be compared with those of other studies.   

Response3: A discussion section has been presented, please review line 425-464.

Round 2

Reviewer 2 Report

The authors changed the title instead of adding the sustainability aspect. As the journal focusses on sustainability I would rather suggest the authors indicate where they can identify it.

Author Response

Point 1: The authors changed the title instead of adding the sustainability aspect. As the journal focuses on sustainability.

I would rather suggest the author indicate where they can identify it.

Response 1: My reply is shown in red words, please view those words in red.
